

# Systematic review and meta-analysis of 50 years of coral disease research visualized through the scope of network theory

Luis M. Montilla, Alfredo Ascanio, Alejandra Verde and Aldo Croquer

Universidad Simón Bolívar, Caracas, Venezuela

## ABSTRACT

Coral disease research encompasses five decades of undeniable progress. Since the first descriptions of anomalous signs, we have come to understand multiple processes and environmental drivers that interact with coral pathologies. In order to gain a better insight into the knowledge we already have, we explored how key topics in coral disease research have been related to each other using network analysis. We reviewed 719 papers and conference proceedings published from 1965 to 2017. From each study, four elements determined our network nodes: (1) studied disease(s); (2) host genus; (3) marine ecoregion(s) associated with the study site; and (4) research objectives. Basic properties of this network confirmed that there is a set of specific topics comprising the majority of research. The top five diseases, genera, and ecoregions studied accounted for over 48% of the research effort in all cases. The community structure analysis identified 15 clusters of topics with different degrees of overlap among them. These clusters represent the typical sets of elements that appear together for a given study. Our results show that while some coral diseases have been studied considering multiple aspects, the overall trend is for most diseases to be understood under a limited range of approaches, e.g., bacterial assemblages have been considerably studied in Yellow and Black band diseases while immune response has been better examined for the aspergillosis-*Gorgonia* system. Thus, our challenge in the near future is to identify and resolve potential gaps in order to achieve a more comprehensive progress on coral disease research.

## INTRODUCTION

Coral diseases have been an important factor responsible for the decline of coral reefs in the last decades (*Rogers & Miller, 2013*). Although pathogens and diseases are part of the natural dynamics of ecosystems, including coral reefs, the interaction with other stressful environmental factors aggravates their negative effects (*Ban, Graham & Connolly, 2014*), enhancing important losses of live coral cover (*Lewis et al., 2017*; *Precht et al., 2016*; *Randall & Van Woesik, 2015*). This loss of coral cover has been particularly important in the Caribbean, which has been frequently called a "coral disease hot spot" because of the number of diseases and the range of affected species in comparison with the low coral cover in the region (*Bruckner, 2002*; *Green & Bruckner, 2000*).

Corresponding author
Luis M. Montilla,
luismmontilla@usb.ve

The evolution of this body of knowledge has been compiled and updated frequently in multiple narrative reviews (*Goreau et al., 1998*; *Bourne et al., 2009*; *Antonius, 1981*; *Green & Bruckner, 2000*; *Richardson, 1998*; *Rosenberg & Ben-haim, 2002*; *Santavy & Peters, 1997*; *Weil, Smith & Gil-Agudelo, 2006*; *Woodley et al., 2015*). However, quantitative approaches (e.g., meta-analyses, systematic reviews) that complement these publications have been less frequent. One example is the result provided by *Work & Meteyer (2014)*, who examined almost 500 coral disease papers, and classified the used methods in terms of six broad categories. From their analysis they evidenced the scarcity of microscopical sign descriptions using histological techniques. In another example, *Ward & Lafferty (2004)* analyzed the frequency of coral disease papers as a potential proxy of a coral diseases incidence. Finally, *Ban, Graham & Connolly (2014)* reviewed the experimental research about at least two stressors simultaneously using network theory. These types of syntheses, although scarce, provide important benefits over narrative reviews such as offering a wide and more objective perspective of the research landscape, integrating the trends in subfields of a discipline, and potentially identifying research gaps and opportunities for new questions to be explored (*Lortie, 2014*).

Here we present a systematic review aimed at identifying groups of coral disease research topics that have been explored more frequently than others. To do this, we performed a network analysis approach. Network analysis is frequently used in systematic reviews (*Borrett, Moody & Edelmann, 2014*; *Ohniwa, Hibino & Takeyasu, 2010*) and allows researchers to address multiple issues in coral disease research, such as the evaluation of phage-bacteria interactions (*Soffer, Zaneveld & Vega-Thurber, 2014*) synergistic effects of environmental stressors (*Ban, Graham & Connolly, 2014*), the analysis of microbial positive and negative interactions in healthy and diseased conditions (*Sweet & Bulling, 2017*; *Meyer et al., 2016*), and the analysis of gene expression and regulation (*Wright et al., 2015*). We hypothesized that if the research topics typically addressed in coral research lack uniformity among several key aspects of epizootiology, then a network representing the co-apparition of these topics in coral disease research papers would exhibit a community structure, where the communities (also called clusters) of nodes would represent the different themes that have characterized most of coral disease research in the last 50 years.

## METHODS

### Data acquisition

We reviewed a total of 719 publications spanning a period from 1969 to 2017. We searched for these in the search engines and databases *Google scholar*, *Meta*, and *Peerus*, using combinations of keywords with search modifiers, including "coral disease", "syndrome", "yellow", "black", "purple", "spot", "band", "pox", "dark", "plague", "growth", "trematode", "anomalies", "ciliate", "soft coral", and "aspergillosis". This included peer-reviewed papers and conference proceedings, since the latter would also provide information about research questions explored in a given time and a given location. We excluded thesis, preprints , and book chapters. Additionally, we compared our database with the list of papers analyzed by *Work & Meteyer (2014)* looking for potential omissions.
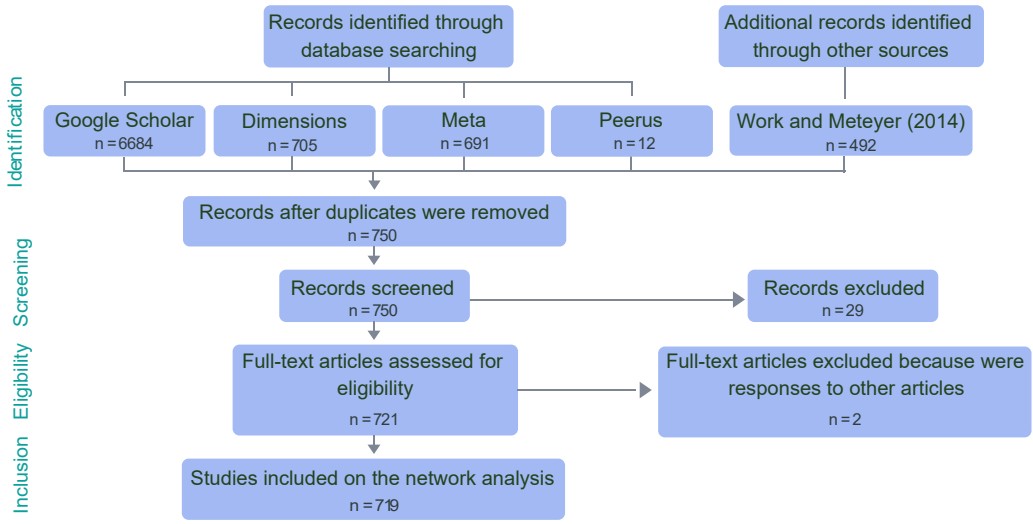

**Figure 1** **PRISMA flow diagram for coral disease papers included in the network analysis.**

The papers were included if the research question directly addressed some biological or ecological aspect of coral diseases or coral pathogens (Fig. 1). The comprehensive list of included papers is provided as Supplemental Information. For every paper, we assessed four fundamental questions, each one corresponding to one node category:

- **What was the studied disease?** Each disease was listed as an individual node and the cases where there was no specific disease of interest (e.g., general surveys), we assigned the node "multiple diseases". Additionally, we classified "White Syndrome" as all the descriptions of pathologies involving tissue discoloration and loss from the Pacific *sensu* (*Bourne et al., 2015*). We applied similar criteria and classified "Pink syndrome" as the references to Pink spots, Pink line syndrome, Pink-Blue syndrome, and Pink-Blue spot syndrome, as these diseases have not been clearly distinguished from one another. We excluded papers specifically concerning thermally-induced bleaching but included the node "bleaching effects" for papers investigating the effects of bleaching over some disease topic or vice versa.

- **Where did the samples come from?** Or **where was the study conducted?** We reported the corresponding marine ecoregion *sensu* (*Spalding et al., 2007*). When no explicit sampling site was stated, we contacted the respective authors to verify the location.

- **What was the genus of the affected host?** If the study implicated multiple specific genera, each one was listed as an individual node. In the cases of baselines and similar studies which lacked specific taxa of interest, we assigned the node "multiple genera".

- **What were the objectives of the study?** We specifically looked into the questions or specific objectives addressed in the introduction section of the papers and assigned a set of keywords. Each keyword represented one node in the network. The complete list of keywords (Supp. 2) is included as a supplementary file at https://github.com/luismmontilla/CoDiRNet/tree/master/supplementary.

Two of the authors (see author contributions) performed the assignation of nodes to each study, and in the case of disagreements, all the authors were consulted until a consensus was reached.

### Network construction and community structure analysis

Using the extracted topics as vertices and their co-occurrence in a given paper as the edges, we built an undirected weighted network. The weight $w_{ij}$ was the frequency of co-occurrence of the keywords $i$ and $j$ in the same article (Fig. 2). The network was constructed and analyzed using the R package *igraph* (*Csardi & Nepusz, 2006*; *RCoreteam, 2016*). Considering the size and complexity of the resulting graph, we used the network reduction algorithm proposed by *Serrano, Boguna & Vespignani (2009)*, implemented in the package *disparityfilter* for R (*Bessi, 2015*) to extract the backbone of our network. This resulted in a smaller graph that retains the multiscale properties of the original network.

Next, we used the link communities approach (*Ahn, Bagrow & Lehmann, 2010*) to obtain groups of nodes forming closely connected groups (hereafter, referred as communities), using the *linkcomm* package (*Kalinka & Tomancak, 2011*). With this method overlapping communities may appear, which allows for several nodes to be part of multiple communities (the scripts for the used functions with their modifications are available at https://github.com/luismmontilla/CoDiRNet. We explored the similarity among the obtained communities, representing them as a new network of communities where each community was a node, and the edges had the value of the Jaccard coefficient for the number of shared nodes (hence, communities without shared topics would be disconnected). Additionally, we used the community centrality (*Kalinka & Tomancak, 2011*) as a measure of the importance of each node within their respective communities.

To test the statistical robustness of the obtained communities, we measured the communities assortativity ($r_{com}$) using a modification of the method proposed by (*Shizuka & Farine, 2016*). In this case, this metric measures a proportion of derived replicates (permuting the network edges) that result in the same community structure as the network derived from the original data. The coefficient is a value ranging from $-1$ to $1$, where values closer to $0$ indicate that the community structure in the network of interest differed from the communities obtained in random permutations; values close to $1$ indicate that the permuted networks have similar node communities as the original network; and values close to $-1$ indicate that the communities of the permutation-derived replicates are formed by different nodes in contrast with the original network. We used a modification (hereafter $r_{oc}$) that allows the use of this coefficient for overlapping communities (code available at https://github.com/alspeed09/OverlapCommAssortativity).

### RESULTS

The content of all the 719 coral disease papers yielded a network comprising 302 vertices and 4,184 edges. The strength distribution (i.e., the distribution of the sum of all connections to the nodes) revealed that the coral disease research network, like other natural networks, presented a highly skewed distribution, where most research topics share few connections and a comparatively small group of topics are heavily represented in scientific publications.

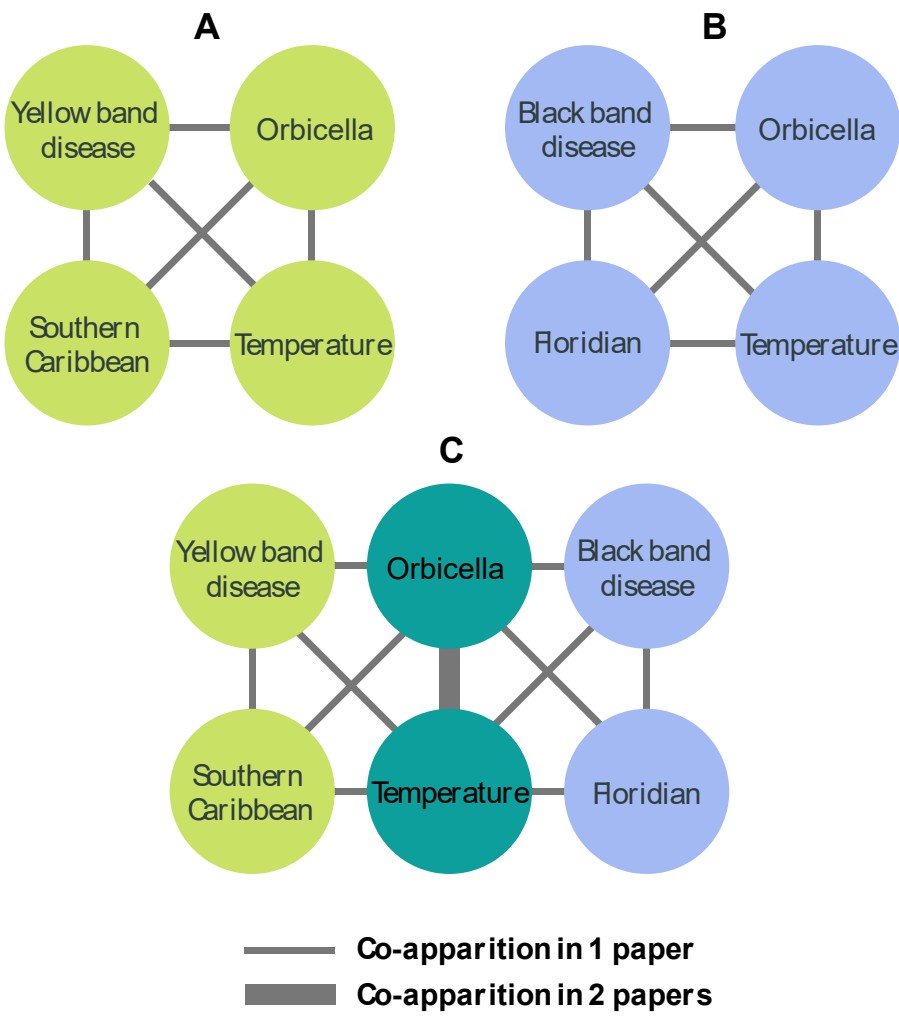

**Figure 2 Network construction.** (A) represents a hypothetical paper focusing on the effect of temperature on of Yellow band disease in a country of the Southern Caribbean. Each paper produces a fully connected graph. In (B) a second hypothetical paper generates its own graph, which has a link between *Orbicella* and Temperature in common with (A). (C) represents the resulting network, with the link between *Orbicella* and Temperature representing the co-apparition in two papers and the remaining links representing one paper.

This pattern persisted for the specific distribution of each type of nodes, suggesting that over the past 50 years, coral disease research has focused on a reduced number of questions in each category (Fig. 3).

Considering this, we explored the identity of the five most represented topics in each case. For all the categories, five topics accounted for over 40% of the total frequency of apparition in all the examined papers (Table 1). We show that the evaluations of prevalence and/or incidence have had a prominent place in research, together with questions about the spatial patterns, transmission, and descriptions of the signs (mostly visual but also histological descriptions) of diseases. The role of temperature stress had also received priority in comparison with other environmental drivers.
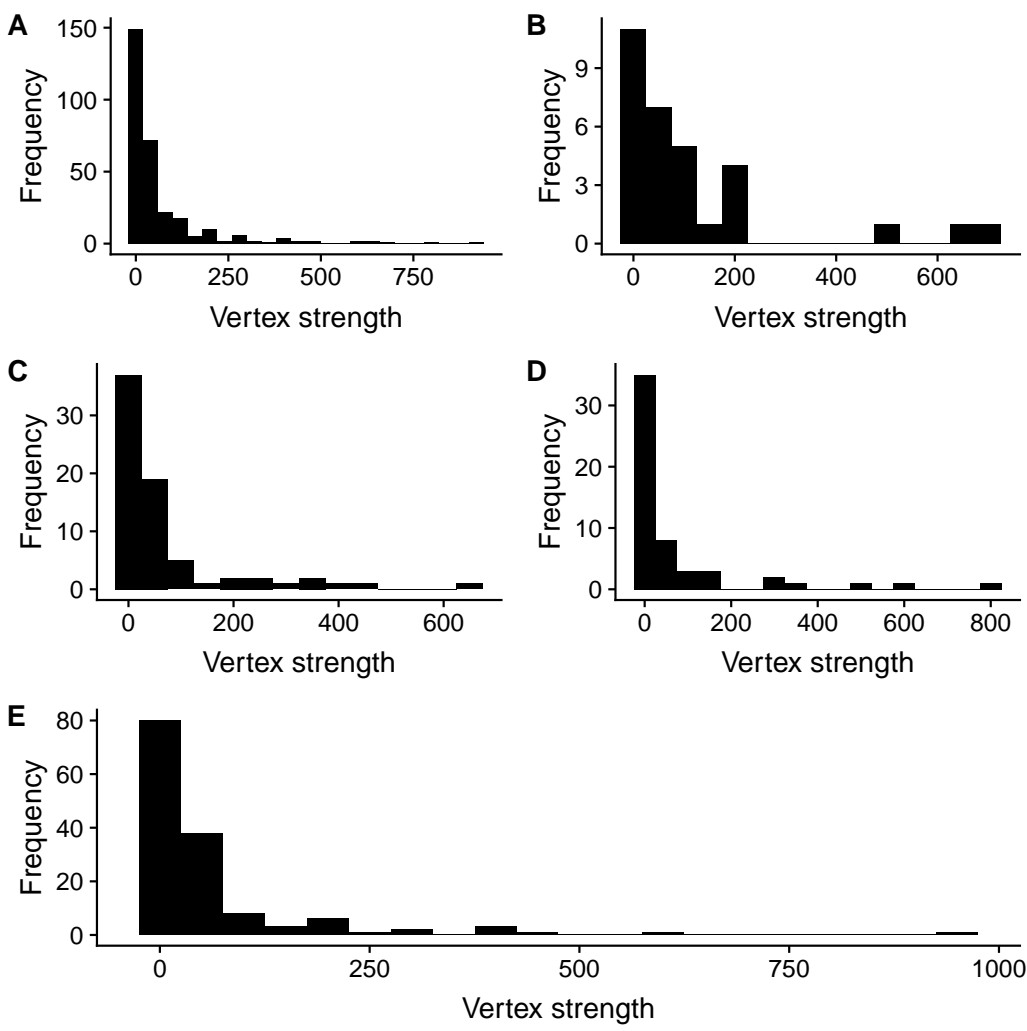

**Figure 3** Strength distributions for **(A)** the global network, **(B)** disease nodes, **(C)** ecoregion nodes, **(D)** genus nodes, and **(E)** objective nodes.

After applying the disparity filter, the initial network was reduced to 93 vertices and 233 edges. The set of excluded research questions were included as supplementary material at https://github.com/luismmontilla/CoDiRNet. These excluded topics either encompass a set of highly-specific questions being scarcely explored (e.g., diseases on mesophotic corals, the effects of ocean acidification, or the role of commensal bacteria) or new questions altogether.

From this reduced network, we obtained a total of 15 node communities (hereafter, referred as C1 to C15). We obtained a $r_{oc}$ value of 0.23, indicating that the network effectively possess a community structure that departs from randomness. These groups ranged from small communities with low overlap, to rich and highly interconnected communities, e.g., C5 and C14 shared almost 50% of their members (Fig. 4).

**Table 1  Top five topics addressed in the coral disease research literature.**

| Category | Topic | Frequency (%) |
|---|---|---|
| Disease | Black band disease | 17.1 |
| | White syndromes | 14.2 |
| | White plague | 6.4 |
| | Yellow band disease | 5.6 |
| | Growth anomalies | 5.4 |
| | **Total** | **48.7** |
| Genus | *Acropora* | 16.3 |
| | *Orbicella* | 12.0 |
| | *Porites* | 9.5 |
| | *Montipora* | 7.6 |
| | *Gorgonia* | 7.4 |
| | **Total** | **52.8** |
| Ecoregion | Floridian | 15.9 |
| | Eastern Caribbean | 9.3 |
| | Greater Antilles | 9.2 |
| | Central and Southern GBR | 8.6 |
| | Hawaii | 7.6 |
| | **Total** | **50.6** |
| Objective | Prevalence/incidence | 23.5 |
| | Spatial patterns | 13.1 |
| | Sign descriptions | 11.8 |
| | Transmission | 10.0 |
| | Temperature | 9.9 |
| | **Total** | **68.3** |

The smaller communities were composed of six or less nodes. C3 represented four topics related to questions about temporal patterns; C6 represented four genera typically affected by Dark Spot disease; C8 was comprised of studies dealing with stress temperature and zooxanthellae damage in the Great Barrier Reef and nearby locations. C9 showed that an important group of studies related to White Band disease have been focused in determining the advance rate, transmission patterns, and usually a description of the signs. C10 included the effects of environmental factors on diseased and bleached corals (Fig. 5).

The medium-sized communities offered more clear pictures of several trends prevailing in the coral disease research. C1 was an *Acropora*-centered community. This included topics related to the descriptions of signs, bacterial assemblages, transmission mechanisms and pattern of several affections like White Syndromes, White Pox, and Brown Band disease, in the Great Barrier Reef. C7 included studies around Caribbean Yellow band disease, mostly about changes in the microbial community, and also temporal and spatial patterns of the disease. C11 was a community constituted by studies about *Vibrio* as a coral pathogen, especially infecting the genera *Oculina*, *Montipora*, and *Pocillopora*. C13 was a community that incorporated two research trends. The most important node in this community was White Syndrome, associated to topics like pathogen characterization, transmission

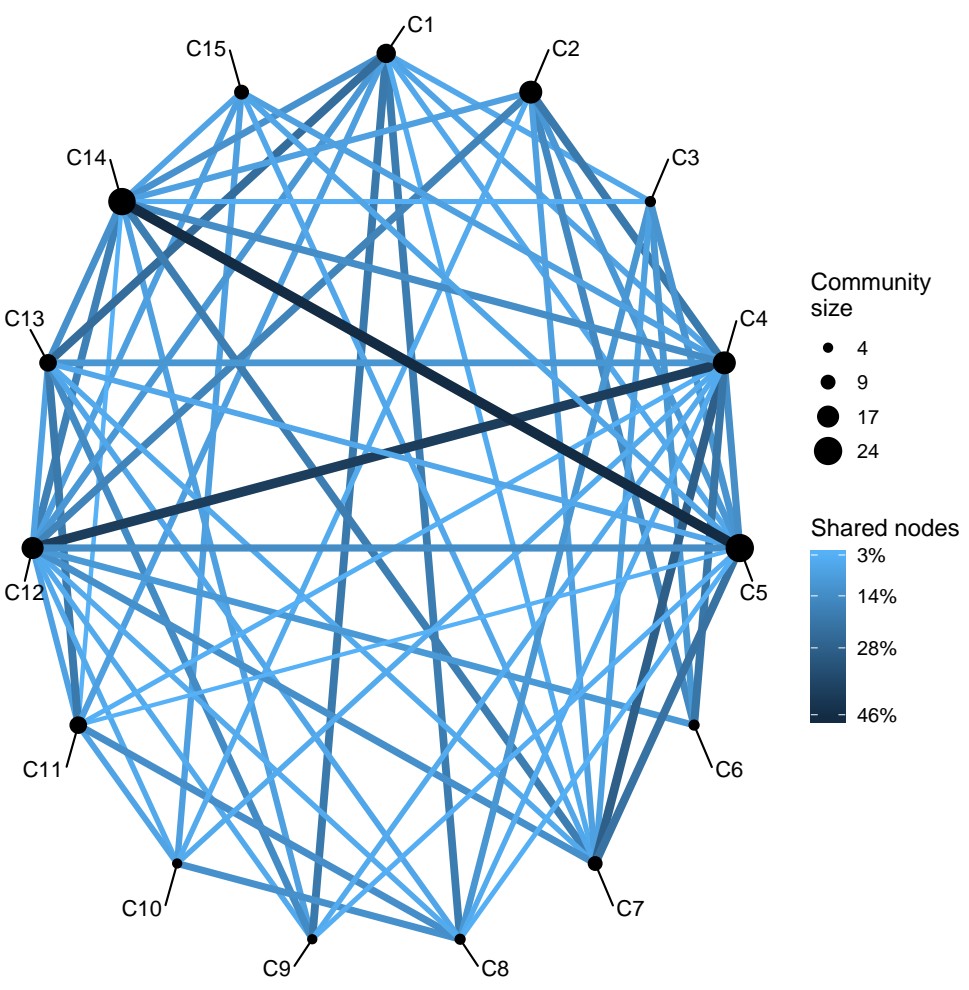

**Figure 4 Similarity network of 15 communities.** Node size represents the number of topics in each community, and edge width represents the similarity between two given communities.

experiments, and changes in bacterial assemblages of the hosts. These hosts were mainly *Montipora* and *Porites*, which were the connection to the second group: research about growth anomalies and trematode infections in Hawaii. C15 was a community centered on gorgonian affections and immune processes (Fig. 6).

All the large communities (nodes > 5) had two nodes in common: Black Band disease and Southern Caribbean. C2 was the most specific one, including objectives related to Black Band disease only. C4 also included Yellow Band disease and White Plague, in this case encompassing studies dealing with transmission experiments, effects of temperature and changes in the associated microbiota. C5 was a community covering reviews, meta-analysis, baseline studies, and outbreaks reports. C12 was very similar to C4, with the distinction of including studies about White Pox, pathogen characterization, particularly *Serratia marscescens*, pathogen metabolites, and the Bahamian ecoregion. C14 consisted of several
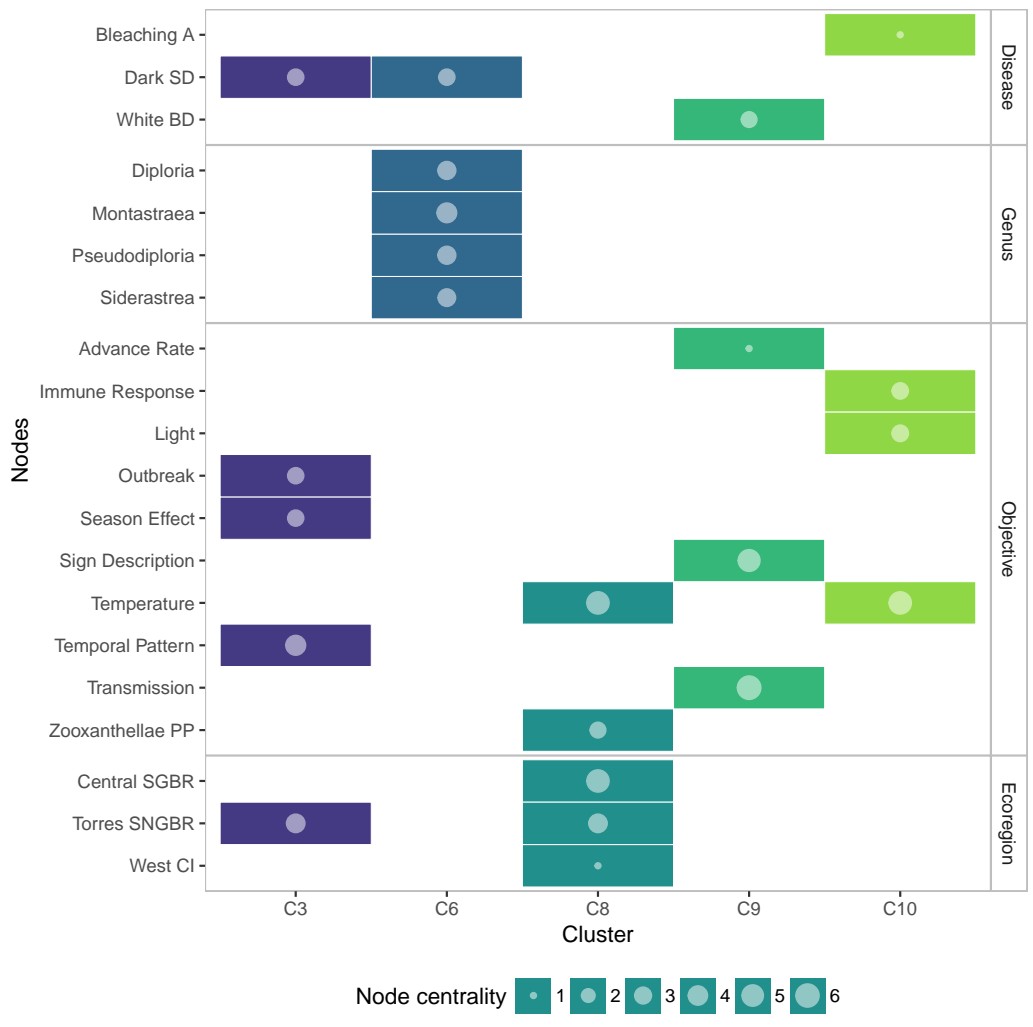

**Figure 5   Membership matrix of communities C3, C6, C8, C9, and C10.** Each column represents a different community and the rows indicate the membership of a given topic to one or multiple communities. e.g., Topics in C9 belong exclusively to this community while the topic Dark Spot disease belong to two different communities. Circle size represents the importance of the topic within a community, measured as overall node centrality.

topics shared with C5, with the particular difference of including studies on *Acropora* and White Syndromes (Fig. 7).

## DISCUSSION

The network approximation used in this study for analyzing the relationships between coral disease research topics provided a quantitative perspective on the intensity of the research and the extension of the field so far. The communities obtained summarized the dominant trends (considering the bulk of the historical research), with each community corresponding to the themes of interest that have accumulated most of the published papers up to 2017. It is important to notice that we expected some minimal community structure

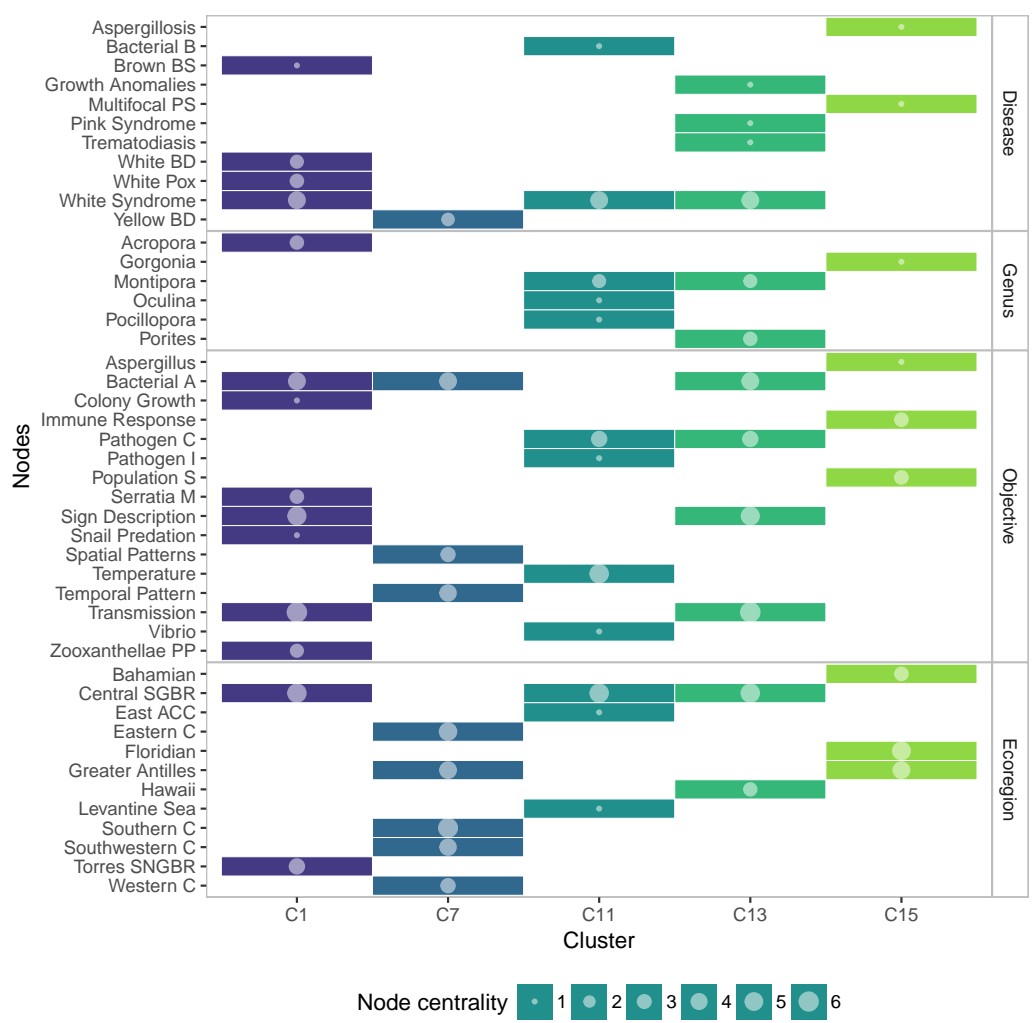

**Figure 6 Membership matrix of communities C1, C7, C11, C13, and C15.** Each column represents a different community and the rows indicate the membership of a given topic to one or multiple communities. Circle size represents the importance of the topic within a community, measured as overall node centrality.

to arise in our network, for some diseases affect only a specific set of coral genera and in certain regions. However, our hypotheses was that the patterns of addressed questions would also contribute to the apparition of multiple communities in our network.

Our results display some clear patterns. For example, Black Band disease occupies a preponderant place in coral disease research as the most connected disease that also appears in most communities with the highest centrality. There are several reasons contributing to this. Black Band disease affection has been studied since the emergence of the field (*Antonius, 1976*; *Garrett & Ducklow, 1975*) and it has consistently accumulated a body of knowledge that makes it the most studied coral disease so far. Its effects comprise a substantial number of coral hosts across different ecoregions and produce extensive mortality during disease epizootics (*Diraviya Raj et al., 2016*; *Aeby et al., 2015*; *Yang et*

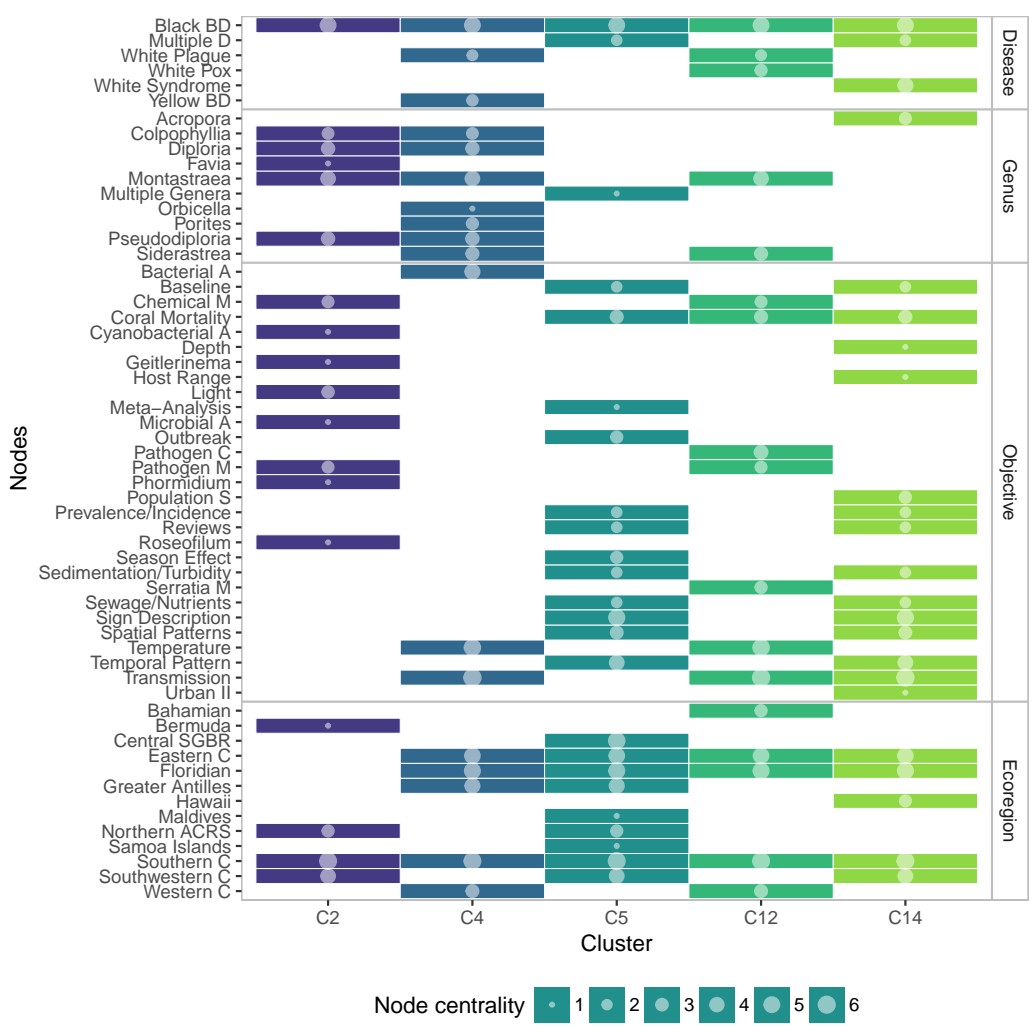

**Figure 7 Membership matrix of communities C2, C4, C5, C12, and C14.** Each column represents a different community and the rows indicate the membership of a given topic to one or multiple communities. Circle size represents the importance of the topic within a community, measured as overall node centrality.

*al., 2014*; *Sato, Bourne & Willis, 2009*; *Hobbs et al., 2015*), especially when combined with seasonal or anomalous temperature increases and determined light conditions (*Chen et al., 2017*; *Lewis et al., 2017*; *Bhedi et al., 2017*; *Miller & Richardson, 2015*; *Kuehl et al., 2011*; *Sato, Bourne & Willis, 2011*; *Boyett, Bourne & Willis, 2007*). Additionally, there have been important advances in the field derived from studies about this disease. Studies about the pathobiome of Black Band disease pioneered the concept of pathogenic consortia as etiological agents of coral diseases (*Carlton & Richardson, 1995*), and the accumulated findings are allowing the proposal of etiological models (*Sato et al., 2016*). In contrast, other diseases have been important enough to appear in their own communities, but they have been associated with less diverse research themes. For example, Caribbean Yellow Band disease has been studied mostly around the changes in its microbial assemblages, and soft

coral Aspergillosis studies usually deal with immune processes; in both cases the number of objectives is similar or lower to the number of ecoregions, implying that research have been extended to different regions but the same set of questions remain.

Our findings also highlight the historical importance of environmental stressors for a broad number of diseases. In our network, temperature was consistently associated with several communities, and the role of sedimentation, turbidity, and nutrient loads (including input through sewage waters) were the main environmental factors associated with baseline studies. This was consistent with a previous review where the authors found that the temperature was the most frequently studied environmental variable in their sample, along with sedimentation, both variables having high influence over coral diseases (*Ban, Graham & Connolly, 2014*).

This contribution complements previous narrative reviews (*sensu Lortie, 2014*; *Petticrew, 2001*; *Morais, Medeiros & Santos, 2018*) about coral diseases; the previously mentioned low-to-mid topic diversity communities together with the list of topics excluded from the backbone network and the list of non-existing links (Supp. 4) can be used as a reference of questions, locations, or affected genera in different coral diseases that could be tackled in future research, especially if we aim to fill existing gaps or strengthen existing but poorly explored processes related to coral diseases.

For example, research about coral immunity has been steadily progressing, however, there is so much room for exploration if we consider that the use of specific approaches can be tested on different coral diseases. For instance, topics like 'gene expression' and 'protein expression' were excluded from the backbone, probably because they represent emergent perspectives, but there are examples of the application of these methods like immune-related transcriptomic profiles that have been developed for Yellow Band disease (*Anderson et al., 2016*), growth anomalies (*Frazier et al., 2017*), and White Band disease (*Libro, Kaluziak & Vollmer, 2013*), while other diseases remain to be explored from this approach. Additionally, the research about coral immunity can be extended when we take into consideration traits like susceptibilities ranges to environmental stress and diseases and the complex symbiotic interactions that constitute the coral holobiont (*Palmer, 2018*), increasing the attention not only to the identity of the microbial assemblage members but also to their functional role in the holobiont. In this sense, in our backbone network, two topics associated to zooxanthellae ('zooxanthellae photosyntetic performance', and 'zooxanthellae damage') were mostly associated to four diseases, and the topic 'functional structure' was excluded from the backbone.

The knowledge about coral disease interactions with environmental stressful conditions can also be widely expanded if we increase the attention to other variables or their interactions with some better studied ones. The role of high temperatures has been the main environmental driver addressed, in coral research, however there are other conditions that have been explored in a few papers like changes in water pH (*Muller et al., 2017*; *Stanić et al., 2011*; *Remily & Richardson, 2006*), dissolved oxygen (*Remily & Richardson, 2006*), and other better studied like the role of nutrient enrichment (*Kaczmarsky & Richardson, 2011*; *Vega-Thurber et al., 2014*; *Voss & Richardson, 2006*; *Bruno et al., 2003*; *Looney, Sutherland*

*& Lipp, 2010*) whose potentially interactive effects with temperature and diseases remain far less explored.

Other relevant aspect of this field that can be improved in the future is the incorporation of open science practices, making available annotated coral disease specific datasets—represented in our network as the 'database' topic—e.g., *Caldwell et al., 2016a*; *Burns et al., 2016*, or the Global Coral Disease Database (https://www.unep-wcmc.org/resources-and-data/global-coral-disease-database), especially considering that we found the topic 'baseline' as an important node in the network but most of this data is not freely available, and it would represent an invaluable resource for further analysis.

In summary, we obtained a generalized representation of the most explored topics in coral disease research. However, these predominant themes and questions are yet to be generalized to the range of potential coral hosts and their diseases. One additional contribution of this work is the availability of a growing database that can be used for further and more diverse exploration of the trends in this field, including the detection of emerging topics and temporal trends, which will help to gain better insights of the development of coral disease research.

## ACKNOWLEDGEMENTS

We wish to thank Laurie Richardson, Longin Kaczmarsky, and Mike Sweet for the information provided. Also Stuart Sandin, Mark Vermeij, Eugenia Sanchez, Bill Gowacki, and Kourtney Barber for their comments on an early manuscript.

### Funding

The authors received no funding for this work.

### Competing Interests

The authors declare there are no competing interests.

### Author Contributions

- Luis M. Montilla conceived and designed the experiments, performed the experiments, analyzed the data, contributed reagents/materials/analysis tools, prepared figures and/or tables, authored or reviewed drafts of the paper, approved the final draft.
- Alfredo Ascanio performed the experiments, analyzed the data, contributed reagents/materials/analysis tools, authored or reviewed drafts of the paper, approved the final draft.
- Alejandra Verde performed the experiments, analyzed the data, contributed reagents/materials/analysis tools, prepared figures and/or tables, authored or reviewed drafts of the paper, approved the final draft, data curation.
- Aldo Croquer authored or reviewed drafts of the paper, approved the final draft.

## Data Availability
The data and the analysis used is available at GitHub:
https://github.com/luismmontilla/CoDiRNet.

## Supplemental Information
Supplemental information for this article can be found online at http://dx.doi.org/10.7717/peerj.7041#supplemental-information.

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
