# Peer review of "Systematic review and meta-analysis of 50 years of coral disease research visualized through the scope of network theory"

_PeerJ, doi:10.7717/peerj.7041_

## Round 0.1 · original submission · Major Revisions

Dear Luis and co-authors,

Please accept my apologies for the delay in response. I was awaiting comments from a third reviewer who is being delayed due to personal matters - these additional comments might be forwarded to you via PeerJ staff in the next few days.

In order not to delay this work any further, here is my recommendation based on two contrasting (one minor and one major) reviews. Both reviewers have acknowledged that the manuscript is of good quality and the findings novel and interesting. However, both reviewers have also noted that the discussion and conclusion sections require additional information, in particular the revised manuscript should provide more emphasis on research gaps and future areas of research.

I will be looking forward to receiving your modified manuscript along with a point-by-point response to the reviewers' comments.

With warm regards,
Xavier

Reviewer 1 ·

Basic reporting

All basic reporting is to standard and no major issues are presented.

Experimental design

The experimental work is well laid out and clear

Validity of the findings

Findings are valid and interesting

Additional comments

Overall;
This study undertakes an assembly of the literature associated with coral disease investigations then conducts a network analysis exercise to identify what diseases have been studied the most and more importantly what maybe understudied and therefore require more attention in further studies.
The approach is interesting and novel and I have no major concerns with the study. It is laid out logically and the construction of the manuscript is solid. It is acceptable for publication with only minor corrections.
The one major element that could be added however in my view - is stronger emphasis on the take home message(s). This approach is most valuable to the scientific field if identifying what are the aspects of coral disease research that should be the focus for future studies. Previous studies have highlighted microscopic histological approaches to link tissue disease signs with the gross lesion characteristics. This current network analysis identified some limitations in the research to date; but does not emphasize strongly enough what should be the future research focus areas. When reading the manuscript these take home messages are just skimmed over. For example linking studies focused on immune systems to some coral disease maybe a recommendation out of the study. Hence I would recommend perhaps in the concluding paragraph a list (perhaps dot point list) of the 4 to 6 focal areas of research that should be undertaken to progress coral disease understanding – that is directly informed from this network theory approach.

Minor Points;

• Abstract line 3: remove “in Order” Start the sentence as “To gain a better insight…
• Introduction Line 2: this sentence should maybe qualified as for some areas; i.e. Not all reefs and regions have been heavily impacted by disease unless you are including Bleaching (which be definition is a disease I know). However in this context I believe the term disease is not referring to bleaching mortality.
• Methods line 48: be sure to make available the papers included in this study; it is says here this is provided as supplementary materials but I could not find this file in the submitted document (unless I missed it).
• Line 89: Agin remove “In Order” not required.
• Table 1: Is the Northern Sector of the GBR separate from the central and Southern regions. Why is the Central and Southern GBR grouped but not the Northern? Just need a justification or some further information on the Bioregions?
• Line 164: infection perhaps instead of “affection”

Reviewer 2 ·

Basic reporting

The paper is well written but the discussion needs to be more fully developed.

Experimental design

This appears to be a standard approach. My only concern is that they define "white syndrome" as tissue discoloration and loss. White syndrome is a tissue loss disease regardless of whether the lesion has discolorations or not. Not sure if that would effect the categorization of their clusters or not.

Validity of the findings

Findings are interesting and valid. However, they need to be presented in a much clearer manner.

Additional comments

This is an interesting approach with the intent to determine where research efforts have been in coral disease research identifying potential gaps. The concept is interesting and I think there is valuable information presented. Table 1 is clear and informative although I would request a full Table with all of the information even as a supplemental file. From the full Table one could better evaluation which corals, regions, etc are completely missing. The authors, however, need a better visual representation of their findings (Fig. 5-7) or perhaps better, more detailed information in the Figure legends. Perhaps something along the lines of Fig. 2? In order for this paper to be useful there needs to be a better explanation of what these “communities” are and what they tell us in terms of research gaps. Discussion needs to be developed more fully explaining the major new understanding of coral disease research efforts. For example, in the discussion it is stated that “For example, Black Band Disease occupies a preponderant place in coral disease research. “ but they don’t explain how the Figures support this. Is it because BBD is linked across 5 clusters (c2, c4, c5, c12, c14)? If so, then they need to spell that out so readers can better understand these results.

---

## Round 0.2 · accepted · Accept

Dear Luis and co-authors,

It is with great pleasure that I accept this manuscript for publication in PeerJ. I have carefully read this final version, and only found a couple very minor edits for you to consider at the proofs stage (see the annotated pdf file attached). This is a very clean, well-written, and comprehensive study that will be of broad interest to the scientific community. Thank you!

With regards,
Xavier

Reviewer 1 ·

Basic reporting

The Revised article is acceptable for publication

Experimental design

Appropriate Experimental Design

Validity of the findings

Appropriate

Additional comments

The revised manuscript is acceptable for publication. No further issues are raised.

Reviewer 2 ·

Basic reporting

Addition of the supplementary materials was really helpful

Experimental design

no comment

Validity of the findings

The authors have done a good job clarifying and expanding upon the relevance of their findings to the field of coral disease research.

Additional comments

This revised manuscript is much more easily understood and will be a nice addition to our knowledge of the state of coral disease research.